# Single nucleus sequencing reveals spermatid chromosome fragmentation as a possible cause of maize haploid induction

Xiang Li[1], Dexuan Meng[2], Shaojiang Chen[2], Haishan Luo[2], Qinghua Zhang[1], Weiwei Jin[2] & Jianbing Yan[1]

Production of maternal haploids using a conspecific haploid inducer is routine and highly efficient in maize. However, the underlying mechanism of haploid induction (HI) is unclear. We develop a method to isolate three nuclei from a pollen grain and four microspores from a tetrad for whole-genome sequencing. A high rate of aneuploidy is observed at the three-nucleus stage (6/22 pollens) rather than at the tetrad stage (1/72 microspores) in one HI line CAU5. Frequent aneuploidy is also observed in another two inducer lines, but not in two regular lines, which implies that HI may be associated with pollen aneuploidy. We further sequence the individual embryos and endosperms of 88 maize kernels crossing between regular and inducer lines. Genome-wide elimination of the CAU5-derived chromosome is identified in eight of 81 embryos. Together, these results suggest that continuous chromosome fragmentation occurring post meiosis in the gametophyte may cause haploidy of the embryo.

[1] National Key Laboratory of Crop Genetic Improvement, Huazhong Agricultural University, Wuhan 430070, China. [2] National Maize Improvement Center of China, Beijing Key Laboratory of Crop Genetic Improvement, Key Laboratory of Crop Heterosis and Utilization, Ministry of Education (MOE), China Agricultural University, Beijing 100193, China. Xiang Li and Dexuan Meng contributed equally to this work. Correspondence and requests for materials should be addressed to W.J. (email: weiweijin@cau.edu.cn) or to J.Y. (email: yjianbing@mail.hzau.edu.cn)

Doubled haploid (DH) technology based on in vivo haploid induction (HI) is used to accelerate the efficiency of breeding in modern maize (Zea mays) improvement programs[1]. Haploids can be obtained by in vitro and in vivo approaches. The most common in vitro approach, male or female gametic cell culture, is often associated with recalcitrance in many species and genotypes[2]. Two in vivo approaches are used to induce haploids in maize. A mutation in the indeterminate gametophyte system, ig, can be used to induce both maternal and paternal haploids, at low frequencies (1–2%)[3, 4]; and inducers derived from the maize Stock6 can lead to maternal haploids[5, 6]; Stock6-derived inducers are used to create maternal haploids in many maize breeding programs worldwide[7, 8]. The first Stock6 inducer line was discovered more than 50 years ago with a HI rate (HIR) of 1–2%[5]. The HI capability of inducer lines can be markedly improved by selection[9], such that the HIR of newly developed inducers, such as CAU5 (an induer line developed at China Agricultural University, ref. [8]), reaches 12%. A number of QTL studies have demonstrated that HI is heritable and controlled by a small number of nuclear genes[10–12]. However, the mechanism underlying Stock6-derived inducer-mediated haploid formation remains unclear. Two hypotheses were proposed to explain this phenomenon. One is the single fertilization hypothesis, in which one of the two sperm cells fails to fuse with an egg cell during double fertilization, and instead triggers haploid embryogenesis, while the other sperm cell may successfully fuse with the central cell, consequently resulting in a functional triploid endosperm[6, 13]. The other hypothesis suggests that two sperm cells may fuse with the egg cell and the central cell, followed by degeneration of inducer-derived chromosomes in the fertilized ovum, which are eliminated in a stepwise fashion from the primordial cells during subsequent cell divisions[14–19]. The cytologic and genetic evidence needed to prove either hypothesis is still largely lacking.

Single-cell sequencing technology emerged recently, enabling high-throughput analyses of the cell lineage trees of higher organisms[20]. In plants, single-cell sequencing is still challenging. We have developed a simple method to isolate and sequence the whole genome of each of the four microspores from a plant tetrad[21]. In the present study, we make further efforts to improve the method for isolating the three nuclei from one mature pollen grain of maize for whole-genome sequencing. Thus, we have the opportunity to directly detect pollen ploidy level at different developmental stages. Combining the single embryo and endosperm sequencing of the hybrid kernel between regular (Zheng58) and inducer (CAU5) lines at 9 days after pollination (DAP), we conclude that chromosome fragmentation starting around the mitotic stage of pollen development is a probable key factor for HI. This study sheds new light on the mechanism underlying maize HI.

## Results

**Pollen fitness varies between inducers and regular lines.** The frequency of kernel abortion in selfed regular inbred lines (3%) was much lower than in selfed inducer lines (10% in CAUHOI and 55% in CAU5) (Supplementary Figs. 1 and 2). In the hybrid ears, nearly 10% kernel abortion was detected when using CAU5 as the male, while almost no aborted kernels were detected when using CAU5 as the female (Supplementary Fig. 1). We thus speculated that the pollen of inducers may be defective compared to regular lines. Pollen competitive ability of inducer lines was weaker than that of regular lines (Supplementary Fig. 3), which agrees with previous results[8]. Pollen fertility in the inducer lines is variable and ranged between 16.4 and 20.9% (Fig. 1a, b; Supplementary Fig. 4a, b), while about 9% aborted pollen was also observed in regular lines, suggesting that pollen fertility may not be the key factor in the induction of haploids. However, significant differences in pollen viability were observed between inducer and inbred lines by 2,3,5-Triphenyltetrazolium chloride (TTC) staining (Fig. 1c, d; Supplementary Fig. 4c, d), and viability could be divided into five classes: high, medium, low, no viability, and abortion. In inducer lines, the proportion of pollen grains with low viability and no viability was higher, and the high viability class was much less abundant than in the regular lines (Fig. 1c, d; Supplementary Fig. 4c, d). The higher proportion of low and non-viable pollen caused by abnormal pollen development may be associated with HI although further evidence is required.

**Single-nucleus sequencing reveals chromosome fragmentation.** Further, we measured the ploidy level of pollen from a inducer line, CAU5. We isolated the three nuclei of a mature pollen grain manually. Each pollen grain released a free nucleus (putative trophic nucleus) and two stick-like compressed nuclei (putative sperm), which were separated with the tip of a glass micropipette under the microscope (Fig. 2a, b). The three nuclei of each pollen grain were placed in individual tubes and lysed for whole-genome amplification, then the genome was sequenced. In total, 66 nuclei from 22 mature pollen grains of CAU5 were isolated, amplified, and whole genome sequenced.

Interestingly, large deletions were observed at a high rate (six among the 22 sequenced pollen grains) of CAU5 (pollens 1–6, Fig. 2c–h). The copy number variations (CNVs) were only detected in the putative sperm, not in the putative trophic nucleus. DNA fragmentation might occur with the trophic nucleus as well; however, it might cause dead or defective pollens and thus not be available for sequencing analysis. Three different types of CNVs were observed: type I, DNA fragments were lost in one putative sperm and gained in the other putative sperm (tan background in Fig. 2c, d); type II, DNA fragments were lost in one putative sperm but not gained in the other putative sperm (light green background in Fig. 2c, e–h); type III, DNA fragments were lost in both putative sperm (dark green background in Fig. 2h). Moreover, the number and distribution of CNVs were different for different pollen grains. CNVs were detected in one or two chromosomes for pollen grains 1–5 (Fig. 2c–g), while CNVs were detected in almost all chromosomes for pollen 6 (Fig. 2h). It seems that the chromosomal deletions might impart variable efficiency and/or initiate at different developmental stages as pollen 6 was more strongly affected, suggesting earlier initiation than in pollen grains 1–5. Interestingly, our results showed that CNVs are concentrated in centromere regions. It is possible that fragments containing centromere satellite sequences can be propagated as minichromosomes instead of being introgressed into the genome[19].

**Fragmentation rate varies between regular and inducer lines.** To determine whether aneuploidy is the underlying reason of HI, the chromosome fragmentation rates in mature pollen of two regular lines B73, Chang7-2 and two inducer lines B73-inducer (~80% B73 and ~20% CAU2 background, CAU2 is an inducer line developed by China Agricultural University, HIR = ~10%), Candidate High Oil Inducer 3 (called CHOI3, HIR = ~7%), were compared (Table 1). For the four lines, we only sequenced the two single-sperm genomes of each pollen (Supplementary Fig. 5), since aneuploidy of the trophic nucleus was not found in the inducer line of CAU5. In total, we isolated 50 B73, 52 B73-inducer, 66 Chang7-2, and 70 CHOI3 sperms from 25, 26, 33, and 35 pollens, respectively, for whole-genome sequencing and CNVs calling.

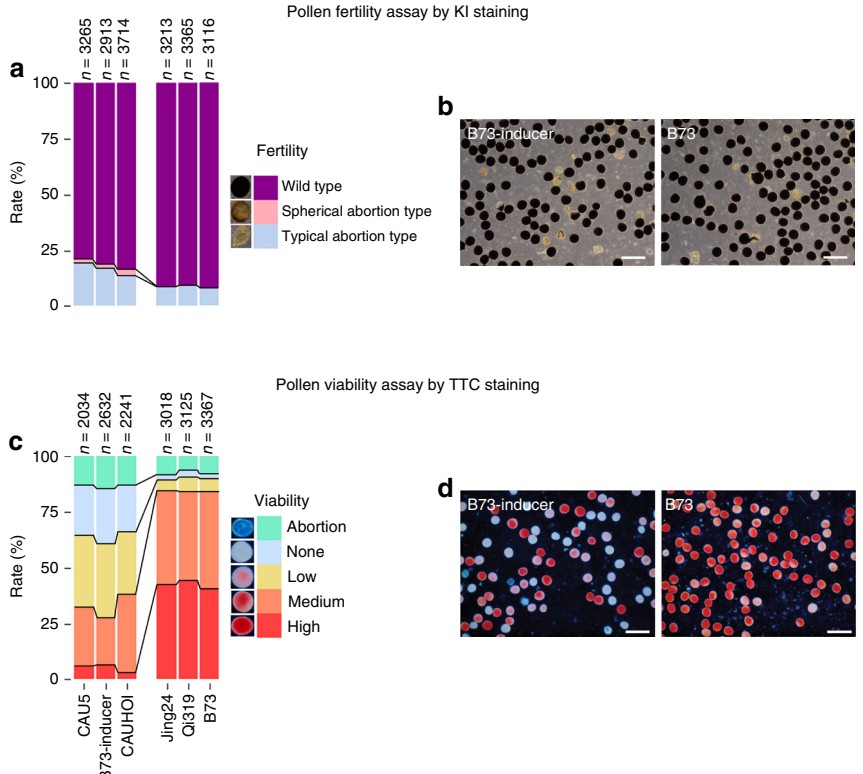

**Fig. 1** Pollen fertility and viability differs between regular and inducer lines. **a** Three levels of pollen fertility, (i) typical abortion type, (ii) spherical abortion type, and (iii) wild type, measured by KI/I$_2$ staining, and counted in the six lines. **b** Pollen fertility of B73-inducer and B73, stained by KI/I$_2$. **c** Five degrees of pollen viability, (i) high, (ii) medium, (iii) low, (iv) no viability, and (v) abortion, measured by the TTC staining and counted in the six lines. **d** pollen viability of B73-inducer and B73, stained by TTC. Scale bar = 1 mm. The fertility and viability photographs and statistic analysis of all six lines are also provided in Supplementary Fig. 4

Only three aneuploid sperms (total 50) were found in B73, significantly fewer ($P = 0.0046$, $\chi^2$ test) than in the B73-inducer (14/52) under the same experimental conditions (Fig. 3a; Supplementary Fig. 5). The difference in aneuploidy frequency between B73-inducer and B73 pollens is marginal ($P = 0.057$, $\chi^2$ test) which could be caused by the limited sample size. In addition, the scale of deletions between B73 and B73-inducer was different. In B73, only few small CNVs were observed in one of the two sperms (B73 pollen2-sperm2, pollen3-sperm2, and pollen18-sperm1, Supplementary Fig. 5); however, in B73-inducer, many large CNVs were observed in most of the chromosomes and five pollens containing two aneuploid sperms were identified (B73-inducer pollens 1, 15, 16, 21, 26, Supplementary Fig. 5), which was similar to the CAU5 pollen 1, 2, 6 (Fig. 2c, d, h).

In Chang7-2, three aneuploid sperms (total 66) were detected, while eight aneuploid sperms (total 70) in CHOI3 were uncovered. Combining the five data sets of single-nucleus sequencing from 2 regular lines and 3 inducer lines, it was clear that the frequency of aneuploidy in the inducer lines is significantly higher than in the regular lines both for pollens ($P = 0.012$, $\chi^2$ test) and for sperm nuleus ($P = 9.5 \times 10^{-4}$, $\chi^2$ test) (Table 1). Compared to the CAU5 aneuploid pollens (except CAU5 pollen 6), the degree of chromosome fragmentation in B73-inducer and CHOI3 seems to be more severe and usually occurred in multiple chromosomes (Supplementary Fig. 5). This suggests that the chromosome fragmentation may be a continuous process since the degree of chromosome fragmentation was different in different pollen stages (CAU5 and the other inducers pollens were sampled before and after anther splitting,

respectively). However, it may be also caused by the genetic background difference of CAU5 and the other inducers, although B73-inducer and CHOI3 shows the same aneuploidy intensity (not frequency), which do not share the same genetic background but were sampled in the same stage.

**Fragmentation initiates around pollen mitosis in HI lines**. Our single pollen nucleus genome sequencing has shed light on the interesting question of when and how the CNVs occur. There are five pollen development stages: microsporocyte, tetrad microspore, one-nucleus pollen, two-nucleus pollen, and three-nucleus pollen, generated by one meiotic and two mitotic processes (Fig. 3b). All six aneuploid pollen grains, which were derived from the same genome at the one nucleus stage, contained at least one euploid nucleus (Fig. 2c–h), suggests aneuploid was induced after the one nucleus stage. Chromosome disruption causing the aneuploidy or CNVs could have been initiated around the 2nd mitosis for type I CNVs (see results above, tan background in Fig. 2c, d), after the 2nd mitosis for type II CNVs (see results above, light green background in Fig. 2c, e, f), and after the first mitosis for type III CNVs (see results above, dark background in Fig. 2h).

To test whether chromosome fragmentation occurs during the meiotic stage, we further sequenced 72 single microspores of 18 tetrads isolated from the HI line for CNV genotyping, based on the isolation method previously described[21]. The analysis method described for mature pollen nucleus sequencing was used. Only one putative CNV was observed in the 72 sequenced microspores (Supplementary Fig. 6), significantly lower than the

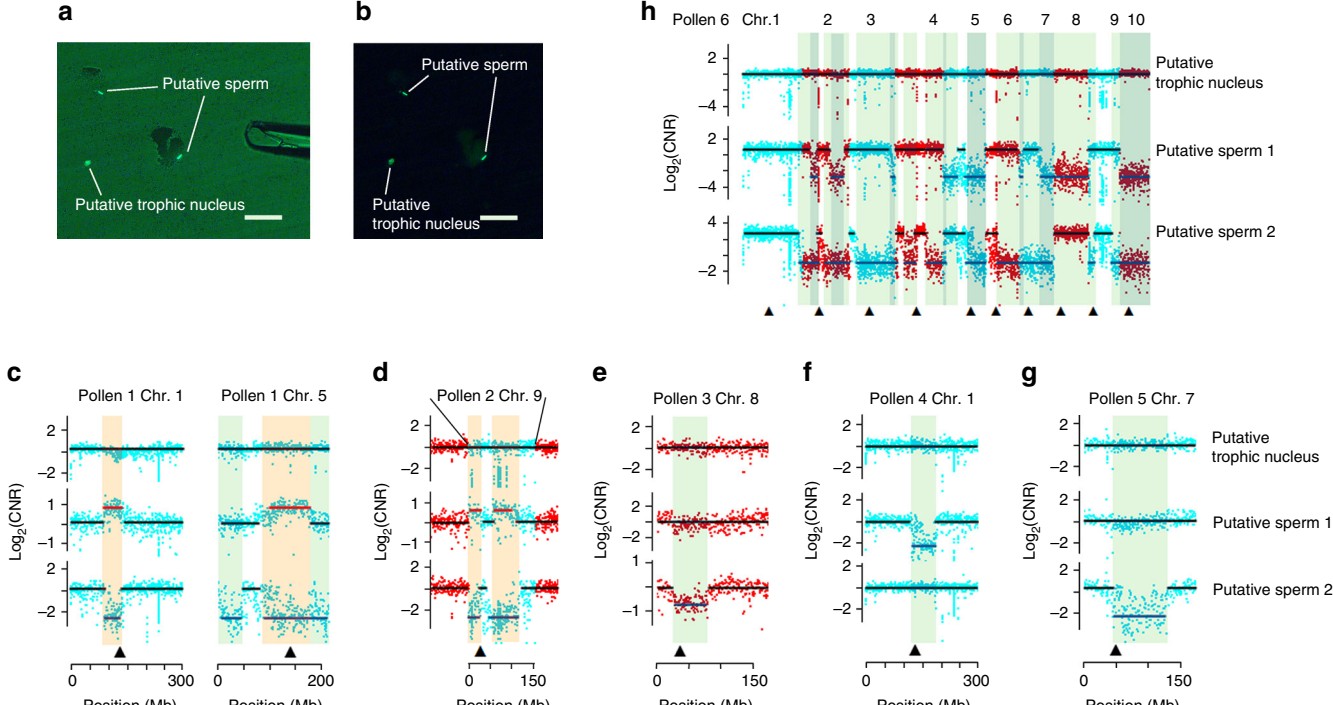

**Fig. 2** Chromosomal fragment deletions identified in inducer CAU5 pollen. **a** Bright illumination plus fluorescence shows pollen cut with the tip of a glass micropipette. **b** One putative trophic nucleus and two putative sperm visualized by fluorescence. In **a**, **b**, DNA is visualized by SYTOX Green fluorescence (bar = 100 μm). **c–g** Pollen grains 1–5 have only one or two defective chromosomes. Some fragments were deleted in one sperm (shown on light green background) or not segregated (deletion in one sperm and diploidy in another sperm, shown on tan color background). **h** Pollen 6 containing two sperm with a large-scale deletions in almost every chromosome. The darker green background shows the deletion regions that occurred in the two putative sperm. In **c–h**, blue and red spots represent CNVs in odd and even chromosomes, respectively; the three data sets are results of putative trophic nucleus, putative sperm 1, and putative sperm 2, from top to bottom; the black, blue, and red lines around spots indicate normal haploidy, deletion, and diploidy, respectively. The black triangles point to centromeres

**Table 1 Aneuploidy frequency of single pollen/sperm sequenced in this study**

| Lines | HIR (%) | AP1S[a] | AP2S[b] | TP[c] | PP[d] | ASS[e] | TS[f] | PS[g] |
|---|---|---|---|---|---|---|---|---|
| B73 | – | 3/25 | 0/25 | 5/58 | | 3/50 | 6/116 | |
| Chang7-2 | – | 1/33 | 1/33 | | | 3/66 | | |
| CAU5 | ~12 | 3/22 | 3/22 | | 0.012 | 9/44 | | $9.5 \times 10^{-4}$ |
| B73-inducer | ~10 | 4/26 | 5/26 | 21/83 | | 14/52 | 31/166 | |
| CHOI3 | ~7 | 4/35 | 2/35 | | | 8/70 | | |

[a]AP1S: amount of pollens containing only one aneuploid sperm/total sequenced pollens
[b]AP2S: amount of pollens containing two aneuploid sperms/total sequenced sperms
[c]TP: the totality of aneuploid pollens/total pollens for regular and inducer lines
[d]PP: $\chi^2$ test P value between regular and inducer lines for pollens
[e]ASS: aneuploid sperms/total sperms
[f]TS: the totality of aneuploid sperms/total sperms for regular and inducer lines
[g]PS: $\chi^2$ test P value between regular and inducer lines for sperms

fragmentation frequency in three-nucleus pollen stage ($P = 0.00005$, $\chi^2$ test, Fig. 3b). The putative small deletion was detected in the peri-centromeric region on chromosome 4 in one of the four microspores of one tetrad. The ratio of CNVs at the tetrad stage (1/18 for tetrads, 1/72 for microspores) was much lower than at the three-nucleus pollen stage (6/22 for pollen, 9/44 for sperm of CAU5; 9/26 for pollen, 14/52 for sperm of B73-inducer; 6/35 for pollen, and 8/70 for sperm of CHOI3). To avoid a biased estimate of aneuploidy due to the limited number of sequenced nuclei, we observed microscopically three aspects of chromosome instability: chromosome deletion, duplication, and lagging during cell division (Supplementary Fig. 7) at the meiotic stage by using three fluorescent markers on chromosomes 2, 4

and 6. These results showed that the rates of aneuploidy at the four meiotic stages (diakinesis, metaphase I, anaphase I, and dyad) were all very low, at <1% in both inducer line CAU5 and regular maize lines Qi319 and Jing24 (Supplementary Data 1). The ratio of chromosome instability in the inducer was slightly higher than in the regular lines, reaching a high of 0.94% at anaphase I for deletions in chromosome 2, but this level is still much lower than the aneuploidy rate of mature pollen grains (6/22 for CAU5, 9/26 for B73-inducer and 6/35 for CHOI3) and on parallel to the rate observed in regular lines during meiosis. Taken together, these results suggest that varying levels of chromosome fragmentation initiate post meiosis in the inducer lines.

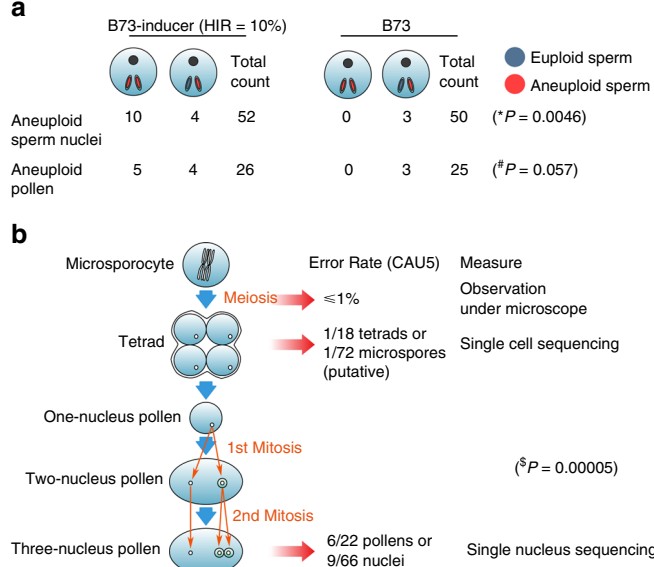

**Fig. 3** Chromosomal fragment deletions in different pollen developmental stages. **a** The number of aneuploid sperm and pollen for B73 and the B73-inducer line. **b** The degree of chromosome fragmentation (in CAU5) was estimated in meiotic and mitotic stage by single cell and nucleus sequencing and observation under microscope. *$\chi^2$ test, $P$ represents the difference of aneuploid frequency of B73-inducer sperms (14/52) and B73 sperms (3/50). #$\chi^2$ test, $P$ represents the significant difference of aneuploid frequency of B73-inducer pollens (9/26) and B73 pollens (3/25). $^\$\chi^2$ test, $P$ represents the difference of aneuploid frequency of CAU5 pollens (6/22) and CAU5 microspores (1/72)

**Sperm DNA fragmentation causes embryo chromosome elimination.** To investigate if aneuploidy at the pollen stage leads to haploid embryos, we also sequenced 81 embryos and 81 endosperms from 88 $F_1$ kernels of a cross between Zheng58 and CAU5 at 9 DAP. Combining the results of CNVs and SNP ratio, most (151/162) of the samples had normal ploidy (Fig. 4a). Two classes of chromosome fragment deletions have been identified in the remaining samples ($n = 11$, Fig. 4). Class I represents genome-wide paternal chromosome elimination and was detected in eight (8/81 = 10%) haploid embryos (Fig. 4b, c) including seven with complete (Fig. 4b) and one with incomplete elimination, in which paternal chromosomal fragments of chromosomes 1, 7, 8, and 9 were still detected (Fig. 4c). This continuous process of chromosome elimination in inducer lines might occur at different stages and with different efficiencies. However, the chromosome elimination in most samples was completed before 9 DAP based on the present data, which agrees with previous results[19]. Class I was only detected in embryo, not in endosperm, owing that perhaps aneuploid endosperm will cause abortion and not be available for sequencing analysis. Class II represented small chromosomal fragment deletions both in the paternal and maternal genomes and were detected in three samples (Fig. 4d–f). In two embryo samples, one paternal deletion (Fig. 4d) and one maternal deletion (Fig. 4e) of class II were observed. In one endosperm sample, maternal deletion of class II was also observed (Fig. 4f). Class I events with a relatively high frequency (8/81 embryos) would probably produce haploid embryos; Class II seems to be a common phenomenon independent of HI and may exist in any genetic background with a low frequency (3/162 in the present study) and is unlikely to cause whole-genome elimination and HI (Fig. 5). Combining all of the results of the pollen and kernel data sets above, we concluded that the continuous chromosome fragmentation initiated post meiosis of pollen development causes the consequent elimination of the paternal genome (Class I), but only those initiated after 2nd mitosis could cause non-aborted kernels with haploid embryos, independently of common minor chromosomal defects such as CNVs detected in the kernels (Class II) (Fig. 5).

## Discussion

DH technology is a key tool for maize breeding today and has contributed to increased efficiency. Understanding the mechanism by which haploids are induced, may help select super-inducer lines and thus enhance breeding efficiency further. In this study, we isolated the four microspores of multiple tetrads and the three nuclei of mature pollen grains of the CAU5 line for whole-genome sequencing. We proposed that a continuous large-scale chromosome fragmentation probably initiates around the mitotic stage of pollen development, and could be the cause of subsequent chromosome elimination during kernel development, which was also supported by comparison of single-sperm sequencing results from two regular lines (B73 and Chang7-2) and two inducer lines (B73-inducer and CHOI3) (Table 1), although we can not absolutely rule out the possibility that the sperm-derived genome may initiate fragmentation during embryonic mitosis to gradually give rise to haploid embryo. Our results provided additional evidence to support the hypothesis of fertilization first followed by chromosome elimination since paternal DNA fragments (Fig. 4c) were still detected in putative haploid embryos at 9 DAP[14–19]. However, we cannot exclude the single fertilization hypothesis as one of the two sperm cells successfully fuse with the central cell and another fails to fuse with the egg cell during fertilizations especially for those sperm with severe DNA fragmentations. Whether the two kinds of hypothesis exist at the same time and which one is the main type still needs further study. Separating the embryo and endosperm from different earlier developmental stages for whole-genome sequencing will provide more clear evidence to answer this question. Based on our results, we proposed a model for HI in which chromosome elimination is not accomplished at one stroke and may be initiated by chromosome defects occurring in the male gamete. The large-scale elimination has been caused in B73-inducer and CHOI3 mature sperms; however, why large-scale elimination was not detected in 81 sequenced endosperms? It could be caused majorly by sampling, we preferred to choose the complete kernels for isolating endosperm and embryo for sequencing. The phenotype of the kernel with diploid endosperm may be different with regular kernel as smaller or abortive. Another possible reason is that the genome elimination in endosperm was slower than in embryo[19], we may not detect it in present study. Haploids will be produced only when sperm which has undergone chromosome fragmentation fertilizes the egg, whereas abortive kernels will be produced if these sperm fertilize the central cell (Fig. 5). We found indirect evidence of this because the frequency of pollens with only one aneuploid sperm (3/22 = 14% in CAU5; 4/26 = 15% in B73-inducer; 4/35 = 11% in CHOI3) was higher than that of the embryo haploidy (8/81 = 10%), and some of the sperm with chromosome fragmentation are expected to fertilize the central cells, causing kernel abortion and thus not entering the observed count. We calculated the abortive kernel frequency in the $F_1$ ears from a cross between regular and inducer lines and found it was similar to the HIR (Supplementary Fig. 1). Thus, the frequency of aborted kernels will increase in selfed inducer lines with high HIR. This was further confirmed by comparing the frequency of aborted kernels in the selfed CAU5 and CAUHOI; the former (55%) was much higher than the latter (11%) (Supplementary Fig. 2), which is one reason the agronomic

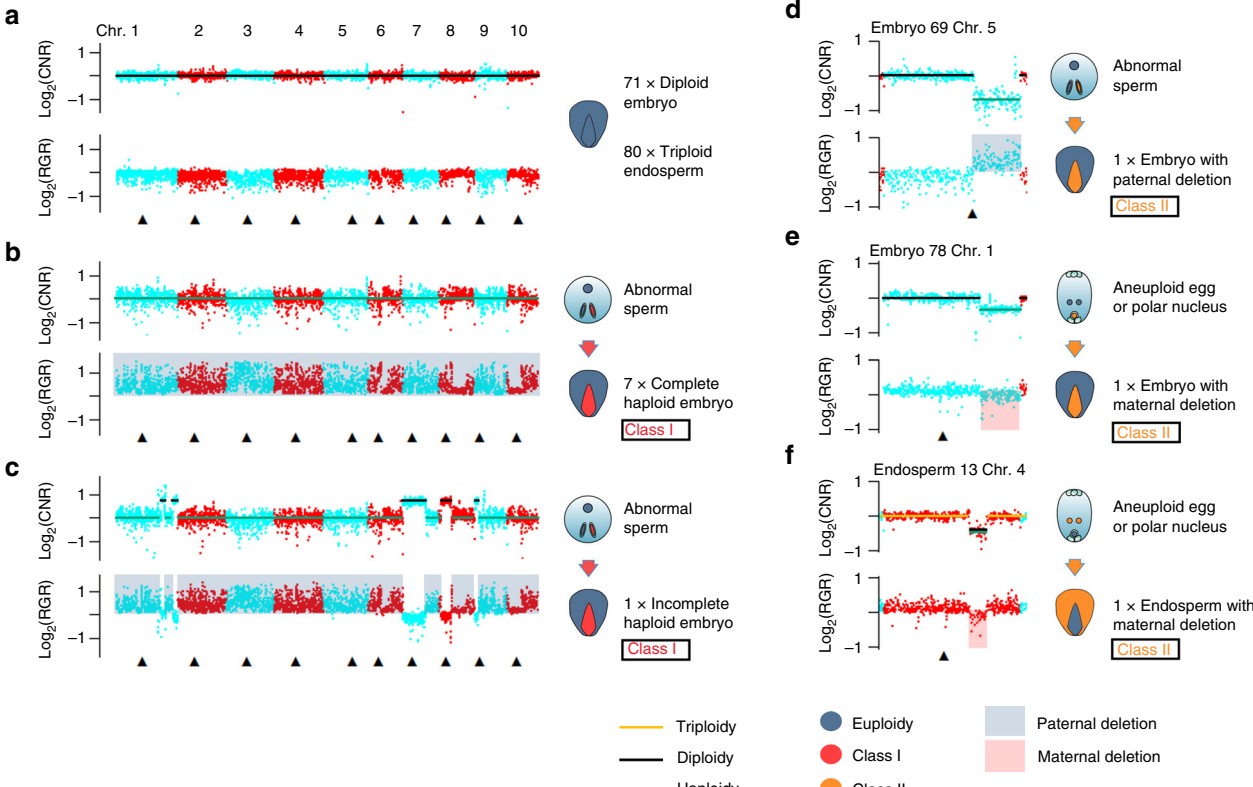

**Fig. 4** Single embryo and endosperm sequencing reveals two classes of aneuploidy. In each panel, the upper dot plot represents CNV by the log$_2$(CNR) value, CNV is identified if continuous and dispersed obviously Log$_2$(CNR) values between large segments (> 10 Mb) are observed. The lower dot plot represents the genetic background SNP-ratio by log$_2$(RGR) value, which means paternal genome deletion if the value is >0, in contrast, maternal genome deletion if the value is <0. The upper and lower figure of each panel is referring to the same sample. **a** The normal euploid embryo (embryo 1). A total of 71 embryos and 80 endosperms were identified as euploid. **b** The complete haploid embryo (embryo 16), in which CAU5-derived chromosomes are eliminated. A total of seven (embryos 16, 19, 21, 32, 38, 45, and 73) were identified as haploid. **c** One incompletely haploid embryo (embryo 44), in which fragments of chromosomes 1, 7, 8, and 9 remain. **d** Chromosome 5 of one embryo (embryo 69), in which the CAU5-derived fragments were lost. **e** Chromosome 1 of one embryo (embryo 78), in which the Zheng58-derived fragments were lost. **f** Chromosome 4 of one endosperm (endosperm 13), in which the Zheng58-derived fragments were lost. The blue and red spots represent CNVs in odd and even chromosomes, respectively. The black triangles point to centromeres. In the CNV plots, the green, black, and yellow lines around spots indicate haploidy, diploidy, and triploidy, respectively. In the SNP-ratio plots, the blue and red shadings indicate paternal and maternal deletions, respectively

performance of inducer lines with high HIR is poor. Chromosome fragmentation is a continuous process initiated post meiotic stage of pollen development, however, only pollens containing only one aneuploid sperm, initiated after 2nd mitosis, could produce non-aborted kernels with haploid embryos. Therefore, increasing the frequency of chromosome fragmentation initiated after 2nd mitosis would be the key to improve HIR, and decreasing chromosome fragmentation initiated before 2nd mitosis would be a efficient way to reduce kernel abortion rate. In wheat[22], Nicotiana[23] and brassica[24], pollen irradiation could also cause parthenogenesis, accompanying with chromosomal rearrangement and fragmentation, which is similar to aneuploidy of maize stock6-derived HI pollens. It suggested that chromosome fragmentation prevents maintenance of the sperm genome in the zygotes.

Improvement of the single-cell sequencing technique has allowed us to isolate the three nuclei from the same pollen[21], which provided us the opportunity to understand the differences in chromosomal abnormality between nuclei and the abnormal chromosome behaviors of the haploid inducer during pollen development. Combining the single microspore sequencing of a tetrad, we could deduce the potential mechanism of HI at the cell level. However, it is still unknown why chromosome CNVs

associated with centromere cause the whole-paternal genome elimination rather than forming partial aneuploids. We found most of the initial CNVs in pollen grains (Fig. 2c–g) were associated with centromere that may the underlying reason of chromosome elimination. A centromeric histone H3 (CENH3) gene was identified in Arabidopsis which could cause uniparental centromere inactivation determining chromosome elimination[25]. However, stock6-derived HI in maize is unlikely to be due to a CENH3 mutation since no homologs were identified in reported HI QTL regions[12]. The major QTL on chromosome 1 affecting HI in maize was cloned by three indepentent groups[26–28] encoding a putative phospholipase A gene named as MTL[26], ZmPLA1[27], and NLD[28]. This gene expresses specifically in pollen developmental stages, is highly conserved across grass species[26–28]. Phospholipase A gene is involved in phospholipid degradation and linolenic acid production and also required for jasmonic acid biosynthesis[29]. However, little is known about the function of this gene especially how it associates with the chromosome stability. The findings in this study will help us to understand the underlying mechanism of HI which is not only contributing to precisely regulating the HIR in maize breeding programs but also to understanding the nature of HI in other species.

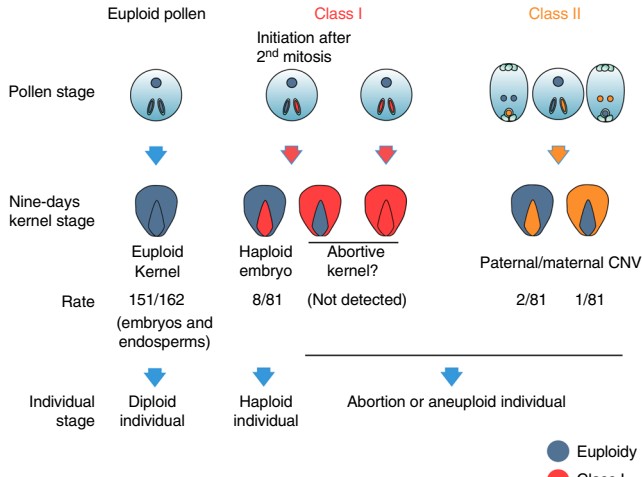

**Fig. 5** A model for haploid generation. Two classes of aneuploid pollen grains were identified. In theory, both classes should carry one aneuploid sperm or two aneuploid sperm, and fertilize egg and/or polar nucleus. Class I pollen grains could induce haploid embryos, genome elimination endosperms, and kernels with approximately equal frequency. Class II pollen grains can fertilize egg and/or polar nucleus and may generate normal or abnormal kernels with a few chromosome fragments lost but not haploid kernels. The frequency of class I is higher than class II based on the present data. Eight haploids with paternal genome elimination were detected in 81 embryo samples. However, no paternal genome elimination was detected in 81 endosperm samples. It is possible that Class I endosperm with genome elimination causes abnormal kernels and may have been ignored when sampling

## Methods

**Maize lines used in present study**. Four inducer lines were used, CAUHOI[16], CAU5[8], B73-inducer[27], and CHOI3[30]. B73-inducer is an introgressed line obtained by crossing CAU2 (an inducer line with HIR = 8%) and B73 followed by two backcrosses with B73 and six selfing generations, resulting in $BC_2F_6$ with ~80% B73 background and 20% CAU2 background. Five regular inbred lines, Qi319 and Jing24, B73, Chang7-2, and Zheng58, were used in this study. When crossing with inducer lines carrying the *R1-nj* marker gene (which imparts a purple scutellum and a "purple crown" of the aleurone, and is widely used for the screening of haploids in maize), an elite hybrid ZD958 (a cross between Zheng58 and Chang7-2) was used as the tester for determining the HIR because of its good purple pigmentation in the endosperm and embryo. The HIR of the three inducers measured in the present study was ~3% (CAUHOI), ~12% (CAU5), and ~10% (B73-inducer), in agreement with previous reports. The HIR of CHOI3[30] was reported as ~7%.

**Kernel classification**. Kernels obtained from self-pollinated and cross-pollinated ears of four lines (CAUHOI, CAU5, Qi319, and Jing24), were classified into three categories according to their morphological features: (i) normal kernels, with normal embryo and endosperm, including normal diploids and haploids; (ii) embryo abortion (EmA) kernels[8, 16, 19]; (iii) endosperm abortion kernels, with shrunken endosperm and kernel size smaller than that of normal diploids (Supplementary Fig. 2). We further divided the endosperm abortion kernels into three classes depending on kernel size and degree of development: (i) abortion in early stage, showed wizened and non-expanded episperm and small size; (ii) abortion in middle stage, with wizened and slightly expanded episperm with small size; (iii) abortion in latter stage, carrying wizened and partly expanded episperm with small size.

**Pollen competitive ability**. Pollen competitive ability is an index reflecting the competitive ability of pollen grains during double fertilizations. The two inducers lines (CAUHOI, CAU5) and the two regular inbred lines (Qi319 and Jing24) were used as pollen sources to evaluate their pollen competitive ability, which was assayed by applying pollen from different lines to the same hybrid ear (ZD958) at different time intervals. The method[8] was described briefly here: pollen from the two inducer lines was applied separately to the different ears of ZD958. After 0, 0.5, 1, 2, 4, and 6 h, the ears pollinated first with the pollen from CAUHOI and CAU5 were

pollinated again with the pollen from Qi319 and Jing24 and vice versa (Supplementary Fig. 3). In order to ensure the uniformity, the newly collected pollen of each line was divided into equal parts through the scale mark of the 1.5 ml tubes. The purple pigmentation of the endosperm can help us to distinguish the two types of pollen-inducer lines or inbred lines. At least 10 ears were used as replicates for each treatment.

**TTC and KI/I₂ staining**. Pollen viability and fertility were measured by TTC staining[31] and KI/I₂ staining[32], compared between the three inducer lines (CAU-HOI, CAU5, and B73-inducer) and the three regular inbred lines (Qi319, Jing24, and B73). First, fresh pollen from the six lines was collected separately at 0900 hours on the same day. Then, the pollen from each line was divided into two samples. One sample was incubated in 1.5 ml centrifuge tube with 0.1% TTC at 37°C for 0.5–1 h, and the other was incubated in 1.5 ml centrifuge tube with 1% KI/I₂ (potassium-iodide/iodine) staining at room temperature for 5 min. The stained pollen was examined and photographed under the microscope. More than 30,000 pollen grains were counted for all lines collected simultaneously, using different days as replicates.

**Probes and fluorescence in situ hybridization (FISH) assay**. The microspores of the meiotic stage of the inducer line (CAU5) and the two regular inbred lines (Qi319, Jing24) were measured for chromosome behavior (Supplementary Fig. 7). The plasmids harboring maize 45 S rDNA tandem repeat[33], maize 5 S rDNA tandem repeat[33] and maize chromosome 4 centromere-specific sequence Cent4[34], were purified and labeled with digoxigenin-11-dUTP and biotin-11-Dutp via nick translation[35]. The digoxigenin- and biotin- labeled probes were detected by anti-digoxigenin antibody conjugated with Rhodamin and anti-avidin antibody conjugated with fluorescein isothiocyanate. FISH images were captured digitally using a CCD camera (QImaging; RETGA-SRV FAST 1394) attached to an Olympus BX61 epifluorescence microscope.

**Single pollen nucleus isolation**. In this study, three parts of single cells or nuclei need to be isolated, such as microspores of CAU5, pollen nuclei of CAU5, and sperm of B73, B73-inducer, Chang7-2, and CHOI3. The haploid inducers and the regular lines were grown in the field. To isolate single tetrad-stage microspores of CAU5, the young tassels, were prepared under the method published previously[21]. We isolated 72 single microspores from 18 tetrads.

To isolate single pollen nuclei of CAU5, the tassels with non-split mature anthers were primarily fixed in Carnoy's Fluid (ethanol: acetic acid = 3:1) for 24 h, then washed with 85% ethanol to replace acetic acid for 4 h, and stored in 75% ethanol. Fixed mature anthers were transferred into a 100 μl drop of 10 μM SYTOX Green (Life Technologies, Eugene, USA) on a slide, and crushed to release pollen, taking care not to fragment the anther wall, as the anther wall nuclei would interfere with observation and isolation of pollen nuclei. After 15 min, the three nuclei could be observed in a pollen grain with SYTOX Green fluorescence under × 10 microscope (Fig. 2a, b). Putative trophic or sperm nuclei could be identified by their shape (the trophic nucleus is spherical and sperm nucleus is stick-like) and validated by single-nucleus sequencing. The two samples (one with chromosome deletions and the other with the complementary gains) should correspond to the two sperm nuclei (Fig. 2c, d). Based on the green fluorescence, the complete nuclei could be separated by using the tip of a glass micropipette as a knife. Each pollen fragment encompassing a single nucleus was then transferred into a plate well with 9 μl water (Sigma-Aldrich, St. Louis, USA). In total, we successfully isolated 66 single nuclei from 22 pollen grains of CAU5.

To isolate sperm of B73, B73-inducer, Chang7-2, and CHOI3, the fresh mature pollens were collected from split anthers and directly transferred into a 100 μl drop of 10 μM SYTOX Green (Life Technologies) on a slide. In this case the three nuclei could be released immediately from the pollen by osmotic pressure, unlike with CAU5. However, the shape of released trophic (stretched) and sperm (round) nucleus are different (Supplementary Fig. 5a). After 5 min, we extracted two single-sperm nuclei to two individual wells with 9 μl water (Sigma-Aldrich, St. Louis, US), using on SYTOX Green fluorescence as a guide. In total, we successfully isolated 50, 52, 66, and 70 single-sperm nuclei from 25 B73, 26 B73-inducer 33 Chang7-2 and 35 CHOI3 pollen grains, respectively.

**Whole-genome amplification and sequencing**. A total of 138 CAU5 samples, including 72 microspores and 66 pollen nuclei, were subjected to the following operations: the GenomePlex Single Cell Whole Genome Amplification Kit (Sigma-Aldrich, St. Louis, USA) was used for nucleus or cell lysis, DNA fragmenting, and polymerase chain reaction (PCR) for whole-genome amplification (WGA) according to the manufacturer's standard protocol. Libraries were made using the TruSeq DNA PCR-Free Sample Preparation Kit (Illumina, San Diego, USA). Whole-genome sequencing for single end was completed in Hiseq 2000 platform (Illumina). For the 238 sperm samples from B73, B73-inducer, Chang7-2, and CHOI3, we used a simpler method to make the libraries, by using the TruePrep DNA Library Prep Kit V2 for Illumina (50 ng for TD501, Vazyme Biotech, Nanjing, China). Pair-end whole-genome sequencing was completed in Hiseq 3000 platform (Illumina).

**CNV calling from single cell and nucleus sequencing data**. Low-quality bases and reads were removed by Trimmomatic 3.0[36]. The remaining high-quality reads were mapped to the maize reference genome AGPv3.18 using the Burrows-Wheeler Alignment tool (BWA)[37]. For CAU5, B73, B73-inducer, Chang7-2, and CHOI3, an total of 0.5, 2.3, 2.6, 3.3, and 2.7 billion filtered reads were obtained per sample; CNVs were called based on the total 151, 127, 162, 147, and 72 million high-quality reads (MapQ > = 10, Supplementary Data 2–6). Two methods are usually used for CNV calling, one calculates the reads density within uniform-fixed-length bins[38–42], and the other defines the uniform-read-depth in the variable bins[43]. In the present study, the 2nd method was used for CNV calling. First, the B73 reference genome was divided into 4000 bins with an average window size of 500 kb, and the average read depth was calculated as 38 k (CAU5), 31 K (B73), 40 K (B73-inducer), 37 K (Chang7-2), and 18 K (CHOI3) over all samples in each bin. Thus, we took the 38 k (CAU5), 30 K (B73), 30 K (B73-inducer), 37 K (Chang7-2), and 18 K (CHOI3) depth as the standards to redefine the bins. Finally, 4005 (for CAU5, Supplementary Data 7), 4315 (for B73, Supplementary Data 8), 5402 (for B73-inducer, Supplementary Data 9), 3990 (for Chang7-2, Supplementary Data 10), and 3991 (for CHOI3, Supplementary Data 11) bins with variable bin length but uniform read depth were obtained. Reads were counted in every bin for every sample. Since read numbers per sample varied, to avoid bias we used copy number of relative reads depth (CNR), which was defined as total counts divided by the median read number, to represent the CNVs for each sample. The CNR value would be around 1. The CNR values were further normalized by Log2 transformation to facilitate chromosome ploidy estimation[42]. Through Log2 transformation, the most of numbers would be calculated as around 0. If it is not euploidy, many constant numbers far away from 0, would exist in some regions. The regions of differing ploidy were indicated by lines (Fig. 2; Supplementary Fig. 5).

**Single embryo and endosperm sequencing**. A total of 81 embryos and 81 endosperms were isolated from 88 hybrid kernels, obtained by crossing inbred line Zheng58 (maternal) and inducer line CAU5 (paternal). DNA of the 9 DAP embryos, endosperms, and the parental leaves were extracted. According to the amount of sample DNA, TruePrep DNA Library Prep Kit V2 for Illumina of different initial DNA amounts (50 ng for TD501, 5 ng for TD502, 1 ng for TD503, Vazyme Biotech, Nanjing, China) were chosen. Kernel DNA libraries were sequenced in 100 bp pair-end of Illumina Hiseq 2000 platform, and paternal DNA was sequenced in 150 bp pair-end of illumina Hiseq 3000 platform.

**CNV calling for single embryo and endosperm data sets**. Removal of low-quality bases and reads and mapping of filtered reads were done following the methods detailed above for single cell and nucleus data analysis. In total, 2.1 billion filtered reads were obtained and 1.9 billion reads were mapped for SNP calling. CNVs were called based on 980 million high-quality reads (MapQ ≥ 10, Supplementary Data 12) on average for each sample. We re-defined the uniform-read-depth bins with the standards described above for single cell and nucleus data since the read amounts and the hot spots of amplification for single embryo and endosperm were different between the single cell and nucleus data set. Finally, we took a 250 k read depth as a standard and 3923 variable-length bins were defined with an average length as 525 Kb (Supplementary Data 13).

**SNP-ratio calling for single embryo and endosperm data sets**. To validate the CNV calling, we used SNP ratio to determine the parental origin of the lost DNA fragments. The hybrid embryo and endosperm should be heterozygous for any SNPs polymorphic between parents. We divided the genome into bins and counted the SNP ratio of each bin. If the genome in a given bin was heterozygous, containing both parents' genomes, the SNP ratio should be close to 0.5 for embryo (0.66 for endosperm). In contrast, if the genome of the given bin was homozygous, i.e., only containing one parent genome, the SNP ratio would be far from 0.5 (or 0.66). Considering that only low sequencing depth (average 0.58×) data were obtained, many false positive SNPs could be generated, resulting in incorrect ploidy assignment. A relative genotype ratio (RGR) instead of the actual SNP ratio in each bin was used. For each bin, the RGR was calculated as the SNP ratio of each sample divided by the average ratio of all the samples, then subjected to log2 transformation.

Reads of paternal (CAU5, depth = 15.6×) and maternal (Zheng58, depth = 16.3×) lines were filtered and mapped with the same standard as described above and 2.9 million high-quality SNPs between parents were obtained. SNPs were called for each embryo and endosperm sample based on the average 11.7 million mapped reads. Average 640 k SNPs for each sample were obtained. There were 711 SNPs on average in each bin, if we defined SNP-ratio bins with 500 Kb in length. In practice, we defined 700 adjacent SNPs as a bin. A total of 4069 diverse length bins with an average length of 503 kb were defined (Supplementary Data 14) and SNP ratio were counted and compared with the CNV calling results. The genetic backgrounds mean paternal genome deletion if the $\log_2$(RGR) values in continuous segments are >0; in contrast, maternal genome deletion if the values are <0.

**Data availability**. The sequencing data for CAU5 single microspores, CAU5 single nuclei, single embryos, and single endosperms have been deposited in the NCBI

Sequence Read Archive under accession code SRP068905. The data sets for single-sperm sequencing of both B73 and B73-inducer have been deposited under accession code SRP081118. The data sets for single-sperm sequencing of Chang7-2 and CHOI3 have been deposited under accession code SRP104117. The authors declare that all other data supporting the findings of this study are available within the manuscript and its Supplementary files.

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

## Acknowledgements

We thank Dr. Jiming Jiang, Dr. Marilyn Warburton, and Dr. James A. Birchler for their helpful comments. This research was supported by the National Key Research and Development Program of China (2016YFD0101003 and 2016YFD0102003), the National Natural Science Foundation of China (31730064, 31525017 and 31421005), the National Youth Top-notch Talent Support Program, and Huazhong Agricultural University Independent Scientific & Technological Innovation Foundation (2014bs22).

## Author contributions

J.Y. and W.J. designed and supervised this study. X.L. and Q.Z. performed the sequencing-associated experiments. X.L. analyzed the sequencing data. D.M. and H.L. completed the other experiments. S.C. developed the inducer materials. X.L., D.M., W.J. and J.Y. prepared the manuscript.
