## [Peer Review File · Nature Communications]

Reviewers' comments:

Reviewer #1 (Remarks to the Author):

Maize specific Stock-6 genotype mediated in vivo haploid induction is the only known intraspecific genome elimination system that is commercially being exploited for Maize breeding programs. Understanding the molecular basis behind this process will not only help to improve this method further in Maize but also would pave way to extend the process of in vivo haploid production from several other agronomically important crop species and thus would be of broad interest to plant community as a whole. However the mechanism of in vivo haploid production in Stock-6 derived haploid inducers is poorly understood, despite several attempts to understand the molecular basis of this interesting phenomenon. In this study Li et al., have taken a step forward to address the mechanistic basis of this process by employing one of the recent advances in high throughput sequencing technology, the single cell genome sequencing approach.

Isolating single cells from plant tissues for single cell genome sequencing is a technically challenging task. Here, the authors have done a commendable job of isolating individual nuclei from a single pollen grain (which comprises 2 sperm nuclei, 1 vegetative nuclei) of Stock-6 derived haploid inducer and sequenced them individually from several pollen grains. Using the information obtained from this sequencing approach along with whole genome sequencing of the double fertilization products, embryo and endosperm of haploid induction crosses they have provided new insights into the mechanism of Stock 6 mediated in vivo haploid induction. This paper shows the power of single cell genome sequencing in addressing an interesting biological phenomena of uniparental genome elimination in plants.

The salient features of this study are

1. There is significant proportion of pollen abortion in haploid inducer lines compared to wild type, which they have clearly shown, is due to chromosome fragmentation leading to CNVs.
2. Chromosome fragmentation specifically occur during haploid mitotic divisions rather than during meiotic divisions which they have clearly demonstrated using pollen from different developmental stages.
3. Interestingly, only sperm nuclei undergoes chromosome fragmentation whereas trophic (vegetative) nuclei does not undergo fragmentation
4. Single embryo and endosperm sequencing indicates that haploids are produced as a result of chromosome fragmentation that takes place during zygotic mitosis till 9 days post fertilization.

The experimental results given in the paper are highly supportive to prove that chromosome fragmentation is occurring in the pollen of haploid inducer but some of the conclusions are hypothetical based on indirect evidences that need some attention as given below.

1. The authors claim that fertilization of egg cell by aneuploid sperm gives rise to haploid embryo is shaky. It may be one possible outcome but there is no direct evidence in support of this claim. At the same time, one cannot rule out the possibility that a normal sperm from an inducer line fertilizes the egg cell and after fertilization, the sperm nuclei then may initiate fragmentation during embryonic mitosis to gradually give rise to haploid embryo. In general, any aneuploid sperm with genetic content less than the haploid genome is less likely to survive, even if it survives it may be outcompeted by the normal wild type sperm during the fertilization process. Even in the recently discovered CENH3 mediated in vivo haploid production, it appears that a normal sperm fertilizes the egg cell and during embryonic mitosis chromosome fragmentation is initiated to produce haploid embryo.

2. It is mentioned that abortive kernels will be produced if an aneuploid sperm fertilizes a central cell

but a haploid embryo if the same fertilizes egg cell. It is not clear from the manuscript why it is so? Endosperm being triploid can tolerate more aneuploidy in general than a diploid. Is there a possibility of obtaining deformed yet viable kernels instead of completely aborted kernels with either haploid embryo or diploid embryo something that resemble category 3 kernels?. It is possible that aneuploid sperm can fertilize endosperm which can induce partial or full genome elimination, yet give rise to viable seeds harboring either diploid / haploid/ aneuploid embryo. This should be elaborated as some of the conclusions in the discussion are based on this observation.

3. It is interesting to note that CNVs are more concentrated in centromere regions. This observation is stated only in discussion but not highlighted anywhere in results section. Is it possible that fragments containing native centromere sequences can be propagated as minichromosomes instead of being introgressed into the genome? Minichromosomes containing centromeres can be stably inherited and thus this may explain why CNVs are concentrated at the centromere regions. Please refer Zhao et al (2013) Plant physiology paper.

Minor suggestions

Is it possible to map some if not all CNVs from the sequencing data for eg. identify few deletions / translocations and use PCR to amplify the junctions and do Sanger sequencing to validate them. This will further support that the CNVs are real and not random artifacts of single nuclei sequencing.

Fig.1 Please include a title text in the main figure itself For eg. " Pollen fertility assay by KI staining" on top in between panels a and b, similarly Pollen viability assay by TTC staining on in between panels c and d. Though each panel is explained in figure legends it is will be less strain for a reader if it is explicitly stated in the figure rather than going back to refer figure legends.

How about rephrasing the class middle as "medium" viability?

Fig.4 : Please label the chromosomes as 1,2 10 either in the top or bottom of the panel

I have noticed following typos in the manuscript

1. Line 232 additional should be additional
2. Line 234 framents should be fragments
3. Line 276 determing should be determining
4. Florescence should be fluorescence (changes required in many places eg. lines 579, 581)

Reviewer #2 (Remarks to the Author):

The manuscript by Li et al. describes novel and interesting observations on maize haploid inducers (HI). Stock 6 was discovered by Ed Coe because it produced haploids when selfed. When used as a pollinator in crosses, it also induces maternal haploids. The mechanism is not understood. The authors decided to characterize the genome of normal vs HI pollen. They separated sperm and vegetative cell nuclei based on differential morphology (collapsed vs round, respectively) and sequenced the single haploid genome. The results are interesting as the haploid inducers display frequent genome instability, manifested by aneuploidy, such as large indels or complete chromosomes. The syndrome is displayed during mitosis of the male gametophyte and affects only the sperms and not the vegetative nucleus or the spore nucleus. The unique behavior of sperms supports the notion that visual identification of the sperm nucleus is working well. The findings, if confirmed, represent a significant step forward in understanding stock 6 haploid induction as well as in elucidating mechanisms for genome surveillance and repair during sexual reproduction. The authors propose a model that is

somewhat (see below) consistent with the evidence. The model is of interest and consistent with our current understanding of seed development. The observations and model potentially explain also why stock 6 generates haploids upon selfing, which is not the case for CENH3-derived HI nor has been described in barley and potato HI systems. Although not mentioned by the authors, pollen irradiation is commonly used to produce "parthenogenetic" embryos reinforcing the proposal that chromosome fragmentation prevents maintenance of the sperm genome in the zygotes.

There are a number of flaws that detract from the work.

1. Although the authors use both HI and non-HI lines in their analysis, the comparison is not extensive. It fails to take full advantage of the B73 and B73 based HI introgression lines. For example, add B73 to Fig. 1. It would be nice to demonstrate that multiple derivatives of Stock 6 share the characteristic genome instability trait and that the frequency of aneuploidy and chromosome breaks (CNV) matches the haploid induction efficiency.
2. The difference between a HI and non-HI strain are quantitative. For example, B73 displays instability as well, albeit at lower rate. The quantitative nature of the trait is not a problem per se, but association of this trait to the HI trait would benefit from including more HI and non-HI lines to verify that the two are tightly associated. Notably, the number of pollen grains with a single affected sperm are close in B73HI and in B73. This is the type of pollen proposed in the model as the most likely to produce haploids. If the model proposed by the authors explains HI, then B73 should be a HI. Some rethinking and revising is appropriate.
3. Statistical tests are missing from data in Fig. 1 and in Fig. 3. In Fig. 1 large numbers are easy to achieve, but the question not addressed is whether different plants with the same genotype (biological reps) would be consistent.
4. Although CNV are obvious, explanation of a systematic procedure for CNV calling is absent. CNR plots seem to be interpreted subjectively by drawing a line from eyeballed trends. The results are arguable in multiple regions such as Fig.2C-pollen1-chr5. At a minimum, a function-dependent smoother should be used. Haploid deletion in a haploid should give only background counts, but they plot at 0.5 CNR. Why? Could a better CNV analysis be carried out? This could be easily done by binning datapoints with a sliding window and comparing across samples with a statistical test. I am also puzzled why the CNR does not display noise on the centromeres.
5. The claim that one stain identifies viability and the other fertility is unsubstantiated and confusing.
6. Procedure for single nucleus isolation is fuzzy and should be better detailed.
7. RGR is calculated by comparison to average ratio of all samples including aneuploids. This could bias analysis. A suggested alternative: compute RGR using euploid controls only. It may be possible to compute a weighted average read count (based on number of reads generated per sample) to account for the wide inter-sample variation in mapped read coverage. Several software packages for calling CNV are publicly available.
8. The table in Fig.3 can be improved by making a column for the totals.
9. Pollen of B73 and B73-inducer is isolated at different stage than CAU5 His. The suggested cause, continuous chromosomal fragmentation, cannot be uncoupled from the effects of the different HI genotypes with the data at hand.
10. The position of cen with respect to CNV is often unclear

10. English:

The article is fairly clear and understandable, but it still needs stylistic and grammar fixes. A few examples

- i. 2nd and second are used the same way.
- ii. fluorescent, introgression are misspelled throughout
- iii. Cross and X are used interchangeably
- iv. Plurals are misspelled.
- v. Superscript3 and (ref. 8) citation style used throughout

Reviewer #3 (Remarks to the Author):

In the present study Li, et al use single cell copy number profiles to determine if aneuploidy could be a cause of haploid induction in maize. They present evidence that viability not fertility is much lower in the pollen of haploid inducing strains. They then show copy number alterations are common in sperm but not trophic nuclei of the pollen of haploid inducing strains, which may or may not be reciprocal between the two sperm. The authors then claim that the fragmentation occurs continuously after mitosis during pollen development. Finally, the authors provide evidence that the paternal genotype is lost after fertilization.

Technically, the use of the GenomePlex kit to amplify genomes of single cells for CNV determination is an appropriate approach. The CNV calling methods used are also standard. However, I do have a few concerns and clarifications needed to paper.

1) Parts of the paper should be rewritten for clarify. For example, lines 160-162 the authors write "It was found that all six aneuploid pollen grains sequenced here contained at least one complete diploid nucleuse, which implied that their origin at the one-nucleus stage muse be euploid, or the aneuploidy initiated during meiosis stage could not be captured since it would cause abortion." Simply stating that all six aneuploid pollen grains, which were derived from the same genome at the one nucleus stage, contained at least one euploid nucleus suggests aneuploid was induced after the one nucleus stage would be much easier to understand.

2) The authors do not provide convincing evidence that the chromosome fragmentation process is continuous. The authors cite the degree of aneuploidy in CAU5 and B73-inducer which were sampled before and after anther splitting. More convincing evidence would be looking at aneuploidy before and after anther slitting in the same strain. The authors look at the tetrad state compared to three nucleus pollen, which provides evidence that the aneuploidy is occurring during the mitotic states. However, it may occur in a single event.

3) The authors do not provide statistical justification for their findings. Biological replicates with statistical tests would strengthen their evidence.

4) Parts of the discussion are irrelevant. For example, the authors spend time discussing CENH3 then conclude it is not likely the culprit because it is not in the HI QTL regions.

5) Figure 4b/c/d are unclear. Figure 4b shows most reads at higher levels than the diploid line the haploid embryo. The sperm in 4c shows several copy number gains. The embryo then shows copy number gain at the sites that were euploid in the sperm and copy number loss at the sites that were gained in the sperm. Similarly in figure 4d the sperm has a large loss in copy number that is gained in the embryo. Can the authors please clarify what is being represented in the figure? A standard copy number plot showing the number of genome copies at each location would be easier to follow, especially when switching between triploid, haploid, and diploid genomes.

Reviewers' comments:

Reviewer #1 (Remarks to the Author):

Maize specific Stock-6 genotype mediated in vivo haploid induction is the only known intraspecific genome elimination system that is commercially being exploited for Maize breeding programs. Understanding the molecular basis behind this process will not only help to improve this method further in Maize but also would pave way to extend the process of in vivo haploid production from several other agronomically important crop species and thus would be of broad interest to plant community as a whole. However the mechanism of in vivo haploid production in Stock-6 derived haploid inducers is poorly understood, despite several attempts to understand the molecular basis of this interesting phenomenon. In this study Li et al., have taken a step forward to address the mechanistic basis of this process by employing one of the recent advances in high throughput sequencing technology, the single cell genome sequencing approach.

Isolating single cells from plant tissues for single cell genome sequencing is a technically challenging task. Here, the authors have done a commendable job of isolating individual nuclei from a single pollen grain (which comprises 2 sperm nuclei, 1 vegetative nuclei) of Stock-6 derived haploid inducer and sequenced them individually from several pollen grains. Using the information obtained from this sequencing approach along with whole genome sequencing of the double fertilization products, embryo and endosperm of haploid induction crosses they have provided new insights into the mechanism of Stock 6 mediated in vivo haploid induction. This paper shows the power of single cell genome sequencing in addressing an interesting biological phenomena of uniparental genome elimination in plants.

The salient features of this study are

1. There is significant proportion of pollen abortion in haploid inducer lines compared to wild type, which they have clearly shown, is due to chromosome fragmentation leading to CNVs.
2. Chromosome fragmentation specifically occur during haploid mitotic divisions rather than during meiotic divisions which they have clearly demonstrated using pollen from different developmental stages.
3. Interestingly, only sperm nuclei undergoes chromosome fragmentation whereas trophic (vegetative) nuclei does not undergo fragmentation
4. Single embryo and endosperm sequencing indicates that haploids are produced as a result of chromosome fragmentation that takes place during zygotic mitosis till 9 days post fertilization.

The experimental results given in the paper are highly supportive to prove that chromosome fragmentation is occurring in the pollen of haploid inducer but some of the conclusions are hypothetical based on indirect evidences that need some attention as given below.

We appreciate the nice summary and the recognitions to our study.

Q1. The authors claim that fertilization of egg cell by aneuploid sperm gives rise to haploid embryo is shaky. It may be one possible outcome but there is no direct evidence in support of

this claim. At the same time, one cannot rule out the possibility that a normal sperm from an inducer line fertilizes the egg cell and after fertilization, the sperm nuclei then may initiate fragmentation during embryonic mitosis to gradually give rise to haploid embryo. In general, any aneuploid sperm with genetic content less than the haploid genome is less likely to survive, even if it survives it may be outcompeted by the normal wild type sperm during the fertilization process. Even in the recently discovered CENH3 mediated in vivo haploid production, it appears that a normal sperm fertilizes the egg cell and during embryonic mitosis chromosome fragmentation is initiated to produce haploid embryo.

A1: Thank you for the nice comments. We agree “we cannot rule out the possibility that a normal sperm from an inducer line fertilizes the egg cell and after fertilization, the sperm nuclei then may initiate fragmentation during embryonic mitosis to gradually give rise to haploid embryo.” However, it is very likely that aneuploidy sperm will cause high rate haploid. And the pollen aneuploidy rate of inducers (6/22 in CAU5; 9/26 in B73-inducer; 6/35 in CHO1₃) was higher than the embryo haploid rate (8/81 in CAU5), which is the indirect evidence of the haploid fragmentation initiation from sperm nuclei stage. It is possible that the serious fragmentation generated in the early stage may not be able to fertilize with the egg cell and generate haploid embryo. Therefore, we summarized that haploid fragmentation initiates from the stage ranging pollen mitosis to embryogenesis. We added “although we cannot absolutely rule out the possibility that a normal sperm-derived genome may initiate fragmentation during embryonic mitosis to gradually give rise to haploid embryo” in the 1st paragraph of Discussions.

Q2. It is mentioned that abortive kernels will be produced if an aneuploid sperm fertilizes a central cell but a haploid embryo if the same fertilizes egg cell. It is not clear from the manuscript why it is so? Endosperm being triploid can tolerate more aneuploidy in general than a diploid. Is there a possibility of obtaining deformed yet viable kernels instead of completely aborted kernels with either haploid embryo or diploid embryo something that resemble category 3 kernels? It is possible that aneuploid sperm can fertilize endosperm which can induce partial or full genome elimination, yet give rise to viable seeds harboring either diploid / haploid/ aneuploid embryo. This should be elaborated as some of the conclusions in the discussion are based on this observation.

A2: We agree that endosperm being triploid can tolerate more aneuploidy in general than a diploid. Why large scale elimination was not detected in 81 sequenced endosperms? It could be caused majorly by sampling; we preferred to choose the complete kernels for isolating endosperm and embryo for sequencing. The phenotype of the kernel with diploid endosperm may be different with regular kernel as smaller or abortive. Another possible reason is the genome elimination in endosperm was slower than in embryo, we may not detect it in present study. In our previous study, the mixoploid were detected in 9DAP, 11DAP, 13DAP, 15DAP endosperm, while in the embryo, mixoploid were only detected in 6-7DAP embryo, but not in 9-10DAP embryo (Zhao et al, 2013, Plant Physiol, 163, 721–731). It

indicated that the process of chromosome elimination in endosperm cells is partial (mosaic) and gradual, and the time window was later in endosperm than that in embryo. We have modified “abortive kernels will be produced...” into “abnormal kernels might be produced” and discussed the underlying reasons.

Q3. It is interesting to note that CNVs are more concentrated in centromere regions. This observation is stated only in discussion but not highlighted anywhere in results section. Is it possible that fragments containing native centromere sequences can be propagated as minichromosomes instead of being introgressed into the genome? Minichromosomes containing centromeres can be stably inherited and thus this may explain why CNVs are concentrated at the centromere regions. Please refer Zhao et al (2013) Plant physiology paper

A3: Thank you for the imaginative comments. If minichromosome really existed which should be a perfect explanation for the mechanism that CNVs are more concentrated in centromere regions. However, at this moment, we have no direct evidence to support it yet. In the new version we added these results in the end of section “Single pollen nucleus sequencing reveals chromosome fragmentation”. “Interestingly, our results showed that CNVs are concentrated in centromere regions especially in type I. It is possible that fragments contain nature centromere sequences can be propagated as minichromosomes instead of being introgressed into the genome¹⁹”.

Minor suggestions

Q4. Is it possible to map some if not all CNVs from the sequencing data for eg. identify few deletions / translocations and use PCR to amplify the junctions and do Sanger sequencing to validate them. This will further support that the CNVs are real and not random artifacts of single nuclei sequencing.

A4: Thank you for the suggestion. However, it is technical difficult, the identify CNVs in present study were Mb-level and it is difficult to define the boundary. And we also have no enough DNA for multiple tests since the single cell amplified by using the GenomePlex Single Cell Whole Genome Amplification Kit (Sigma-Aldrich, St. Louis, US) with limited DNA products. As reviewer 3 mentioned, present single cell sequencing and CNV calling method is following the standard protocol. Some artifacts might exist but the trends should not be changed.

Q5: Fig.1 Please include a title text in the main figure itself For eg. “ Pollen fertility assay by KI staining” on top in between panels a and b, similarly Pollen viability assay by TTC staining on in between panels c and d. Though each panel is explained in figure legends it is will be less strain for a reader if it is explicitly stated in the figure rather than going back to refer figure legends.

A5: Thank you for the suggestion. We have changed it.

Q6: How about rephrasing the class middle as “medium” viability?

A6: Thank you for the suggestion. We have changed it.

Q7: Fig.4 : Please label the chromosomes as 1,2 ... 10 either in the top or bottom of the panel

A7: Thank you for the suggestion. We have changed it.

Q8: I have noticed following typos in the manuscript

1. Line 232 additional should be additional
2. Line 234 fragments should be fragments
3. Line 276 determing should be determining
4. Florescence should be fluorescence (changes required in many places eg. lines 579, 581)

A8: Thank you for the suggestions. We have changed them and also double checked the whole manuscript carefully.

Reviewer #2 (Remarks to the Author):

The manuscript by Li et al. describes novel and interesting observations on maize haploid inducers (HI). Stock 6 was discovered by Ed Coe because it produced haploids when selfed. When used as a pollinator in crosses, it also induces maternal haploids. The mechanism is not understood. The authors decided to characterize the genome of normal vs HI pollen. They separated sperm and vegetative cell nuclei based on differential morphology (collapsed vs round, respectively) and sequenced the single haploid genome. The results are interesting as the haploid inducers display frequent genome instability, manifested by aneuploidy, such as large indels or complete chromosomes. The syndrome is displayed during mitosis of the male gametophyte and affects only the sperms and not the vegetative nucleus or the spore nucleus. The unique behavior of sperms supports the notion that visual identification of the sperm nucleus is working well. The findings, if confirmed, represent a significant step forward in understanding stock 6 haploid induction as well as in elucidating mechanisms for genome surveillance and repair during sexual reproduction. The authors propose a model that is somewhat (see below) consistent with the evidence. The model is of interest and consistent with our current understanding of seed development. The observations and model potentially explain also why stock 6 generates haploids upon selfing, which is not the case for CENH3-derived HI nor has been described in barley and potato HI systems. Although not mentioned by the authors, pollen irradiation is commonly used to produce "parthenogenetic" embryos reinforcing the proposal that chromosome fragmentation prevents maintenance of the sperm genome in the zygotes.

Thank you for the nice comments. The mechanism of pollen irradiation may be similar to it of stock6-derived HI. We added it into the end of Discussion part.

There are a number of flaws that detract from the work.

Q1. Although the authors use both HI and non-HI lines in their analysis, the comparison is not

extensive. It fails to take full advantage of the B73 and B73 based HI introgression lines. For example, add B73 to Fig. 1. It would be nice to demonstrate that multiple derivatives of Stock 6 share the characteristic genome instability trait and that the frequency of aneuploidy and chromosome breaks (CNV) matches the haploid induction efficiency.

A1: Thank you for the suggestions. B73 was added in Fig. 1. And we have also compared a number of regular lines and inducer lines and the detail was shown in a new table (table 1). For sequencing analysis, we added one new regular line (Chang7-2) and one new inducer line(CHOI₃), now we have 3 inducer lines (CAU5, B73-inducer and CHOI₃) and two regular lines (B73 and Chang7-2). The results are consistent. As you know, single cell sequencing is high cost and we hope reviewer appreciate the efforts we have made now.

Q2. The difference between a HI and non-HI strain are quantitative. For example, B73 displays instability as well, albeit at lower rate. The quantitative nature of the trait is not a problem per se, but association of this trait to the HI trait would benefit from including more HI and non-HI lines to verify that the two are tightly associated. Notably, the number of pollen grains with a single affected sperm are close in B73HI and in B73. This is the type of pollen proposed in the model as the most likely to produce haploids. If the model proposed by the authors explains HI, then B73 should be a HI. Some rethinking and revising is appropriate.

A2: Thank you for the suggestion. We decided to take it and added one new regular line (Chang7-2) and one new inducer line(CHOI₃), now we have 3 inducer lines (CAU5, B73-inducer and CHOI₃) and two regular lines (B73 and Chang7-2). Detailed comparisons are shown in Table 1 and we found the results are consistent and significant different statistically. Primarily, it is expected that aneuploidy frequency of B73-inducer sperm nucleus (14/52) and pollen (9/26) is much higher than it of B73. Indeed, there is a low level of aneuploidy in B73 (3/50 sperm and 3/25 pollen). Based on our hypothesis, only the serious fragmentation produced in the later stage of pollen development (Class I, Figure 4-5) will cause HI. The sperm with serious fragmentation in the early stage may not survive or cause abnormal kernel. So a low frequency aneuploidy of B73 sperm could be normal and acceptable. Actually, HIR is a quantitative trait. B73 and other regular lines also have a low capability (perhaps HIR < 0.5%) to induce haploid. In B73, only few small CNVs were observed in one of the two sperms (B73 pollen2-sperm2, pollen3-sperm2, pollen18-sperm1, Supplementary Fig. 5); however, in B73-inducer, many large CNVs were observed in most of the chromosomes and five pollens containing two aneuploid sperm were identified (B73-inducer pollens 1, 15, 16, 21, 26, Supplementary Fig. 5), which was similar to the CAU5 pollen 1, 2, 6 (Fig. 2c, d, h). Similar phenomena were also observed in the other two newly added lines (Chang7-2 and CHOI₃). Aneuploid in regular lines (B73 and Chang7-2) seem be different from the whole genome wide fragmentation of the other aneuploid sperms, and might not cause HI.

Q3. Statistical tests are missing from data in Fig. 1 and in Fig. 3. In Fig. 1 large numbers are easy to

achieve, but the question not addressed is whether different plants with the same genotype (biological reps) would be consistent.

A3: In Fig. 1, as reviewer 2 mentioned, large numbers are easy to achieve. The large number was counted from 10 photo views which were obtained from different individuals and contained ~300 pollens for each photo. Now to address whether different plants with the same genotype were consistent, we added the separated numbers of the three biological repeats and statistics into Supplementary figure 4. The results showed they were consistent.

In Fig. 3, the results were counted from only 25 B73 pollens, 26 B73-inducer pollens, 22 CAU5 pollens and 72 CAU5 microspores. We calculated the P value by χ^2 test with following comparisons. 1) the aneuploidy frequency of CAU5 pollens (6/22) and CAU5 microspores (1/72) is significantly different ($P = 5 \times 10^{-5}$). 2) the aneuploidy frequency of B73-inducer sperms (14/52) and B73 sperms (3/50) is significantly different ($P = 4.6 \times 10^{-3}$). It demonstrated that the sperm fragmentation rate of inducers was higher than regular lines significantly. 3) The aneuploidy frequency of B73-inducer pollens (9/26) and B73 pollens (3/25) is marginally significant ($P = 0.057$), which may be due to the limited amounts. Therefore, we have added the P values into Fig. 3 and the main text. And significant difference was observed if we combined all the data together (detailed was shown in Table 1).

Q4. Although CNV are obvious, explanation of a systematic procedure for CNV calling is absent. CNR plots seem to be interpreted subjectively by drawing a line from eyeballed trends. The results are arguable in multiple regions such as Fig.2C-pollen1-chr5. At a minimum, a function-dependent smoother should be used. Haploid deletion in a haploid should give only background counts, but they plot at 0.5 CNR. Why? Could a better CNV analysis be carried out? This could be easily done by binning datapoints with a sliding window and comparing across samples with a statistical test. I am also puzzled why the CNR does not display noise on the centromeres.

A4: For the CNV calling method, the read number per bin per sample was divided by the median read number for any given sample. The copy number of relative reads depth (CNR) would be around 1. Through Log2 transformation, most of the regions would be calculated as around 0. If it is not euploidy, many constant numbers will be far away from 0 as shown in Figure 2. The method may not be applicable to gene-scale CNV analysis of cancer cells, but must be nice for the large scale (i.e. >10Mb) CNV calling, like in present study. The line was not drawn randomly but calculated based on the median value. For Fig.2C-pollen1-chr5, it was not perfect, however, we can still see the fragments lost. To be honest, we have no strength to develop a new method for CNV calling but borrowed the standard ones as Reviewer 3 mentioned (Knouse et al. PNAS 2014), in which no noise on the centromeres was observed. We defined uniform-read-depth bins to count reads, followed with the method of ref. 38 (Navin, N. et al. Nature 2011); then we calculated the relative depth, and normalize it by Log2 transformation, which followed with the method of

ref. 37 (Knouse et al. PNAS 2014). We also added the detail into methods.

Q5. The claim that one stain identifies viability and the other fertility is unsubstantiated and confusing.

A5: We are using the public known methods for viability and fertility observation. We are sorry not cite the previous studies in the appropriate places. Now, the correct references of pollen viability and fertility by TTC (Duncan et al. Planta 1985) and KI/I₂ (Li et al. Nat. Communications 2013) staining are added.

Q6. Procedure for single nucleus isolation is fuzzy and should be better detailed.

A6: Thanks for the suggestion. We provided detail in the section "Single pollen nucleus isolation" now.

Q7. RGR is calculated by comparison to average ratio of all samples including aneuploids. This could bias analysis. A suggested alternative: compute RGR using euploid controls only. It may be possible to compute a weighted average read count (based on number of reads generated per sample) to account for the wide inter-sample variation in mapped read coverage. Several software packages for calling CNV are publicly available.

A7: Thank you for the nice suggestions. Previously, we calculated the ratio of background-derived SNPs for each bin, then divided by the mean ratio of each sample. Now, we calculated it divided by the mean ratio of the euploid samples. The results are shown as the following figure. In each panel, the above plots are the previous results and the below plots are drawn by the new calculations with the reviewer suggestions. The trends are very similar. At this moment, we didn't change the methods.

Fig. CNV calling methods comparisons.

Q8. The table in Fig. 3 can be improved by making a column for the totals.

A8: Thank you for the suggestion. We are happy to take it and now Fig. 3 looks better.

Q9. Pollen of B73 and B73-inducer is isolated at different stage than CAU5 His. The suggested cause, continuous chromosomal fragmentation, cannot be uncoupled from the effects of the different HI genotypes with the data at hand.

A9: We made the conclusion that chromosome fragmentation is a continuous process based on the data from tetrad stage, three-nucleus stage and immature kernel (9 DAP) of CAU5 genotype majorly. There were two meanings for continuous chromosome fragmentation process. One is chromosome fragmentation initiated post meiosis of pollen development, but the chromosome losing is continuous process which can continue until 9 days after pollination. In present study, we provided the direct evidence for the above mentioned hypothesis. Our previous study also supports it (Zhao et al, 2013, Plant Physiology). Another is chromosome fragmentation initiated in different stages. We have no direct evidence to support whether the chromosome fragmentation initiated in a single event or in different stages. The experimental design suggested by the reviewer 3 will help to test the second hypothesis. In present study, we are referring to the first hypothesis. Of course, we agree, the difference between B73-inducer and CAU5 could be caused by the genetic background. Relevant discussions were added.

Q10. The position of cen with respect to CNV is often unclear

A10: Yes, you are right. We just estimated the positions of centromere based on the B73 reference genome with the resolution in Mega base level.

Q11. English:

The article is fairly clear and understandable, but it still needs stylistic and grammar fixes. A few examples

- i. 2nd and second are used the same way.
- ii. fluorescent, introgression are misspelled throughout
- iii. Cross and X are used interchangeably
- iv. Plurals are misspelled.
- v. Superscript3 and (ref. 8) citation style used throughout

A11: Thank you for your kind suggestions. We have corrected all the mentioned errors and double checked the whole manuscript again. For the citation style, we just follow the request of the nature series journals, the style need to be as (ref. 8) distinguished from the 8th power of the number when the citation is behind the number.

Reviewer #3 (Remarks to the Author):

In the present study Li, et al use single cell copy number profiles to determine if aneuploidy could be a cause of haploid induction in maize. They present evidence that viability not fertility is much lower in the pollen of haploid inducing strains. They then show copy number alterations are common in sperm but not trophic nuclei of the pollen of haploid inducing strains, which may or may not be reciprocal between the two sperm. The authors then claim that the fragmentation occurs continuously after mitosis during pollen development. Finally, the authors provide evidence that the paternal genotype is lost after fertilization.

Technically, the use of the GenomePlex kit to amplify genomes of single cells for CNV determination is an appropriate approach. The CNV calling methods used are also standard. However, I do have a few concerns and clarifications needed to paper.

Thank you for the comments.

Q1. Parts of the paper should be rewritten for clarify. For example, lines 160-162 the authors write "It was found that all six aneuploid pollen grains sequenced here contained at least one complete diploid nucleuse, which implied that their origin at the one-nucleus stage muse be euploid, or the aneuploidy initiated during meiosis stage could not be captured since it would cause abortion." Simply stating that all six aneuploid pollen grains, which were derived from the same genome at the one nucleus stage, contained at least one euploid nucleus suggests aneuploid was induced after the one nucleus stage would be much easier to understand.

A1: Thank you very much for the nice suggestion. We are happy to take it.

Q2. The authors do not provide convincing evidence that the chromosome fragmentation process is continuous. The authors cite the degree of aneuploidy in CAU5 and B73-inducer which were sampled before and after anther splitting. More convincing evidence would be looking at aneuploidy before and after anther slitting in the same strain. The authors look at the tetrad state compared to three nucleus pollen, which provides evidence that the aneuploidy is occurring during the mitotic states. However, it may occur in a single event.

A2: There were two meanings for continuous chromosome fragmentation process. One is chromosome fragmentation initiated post meiosis of pollen development, but the chromosome losing is continuous process which can continue until 9 days after pollination. In present study, we provided the direct evidence for the above mentioned hypothesis. Our previous study also supports it (Zhao et al, 2013, Plant Physiology). Another is chromosome fragmentation initiated in different stages. We have no direct evidence to support whether the chromosome fragmentation initiated in a single event or in different stages. The experimental design suggested by the reviewer will help to test the second hypothesis. In present study, we are referring to the first hypothesis.

Q3. The authors do not provide statistical justification for their findings. Biological replicates with statistical tests would strengthen their evidence.

A3: Thank you for the suggestions. Statistical justification now was added in the appropriate places in Figure 1, Supplementary Figure 4, Figure 3. Biological replicates are a big challenge for genome sequencing study. However, the reviewer raised an interesting question whether the chromosome fragmentation affected by environments. Now we added more regular and inducer lines as reps and significant difference statistically were observed. We summarized the results in a new table (Table 1).

Q4. Parts of the discussion are irrelevant. For example, the authors spend time discussing CENH3 then conclude it is not likely the culprit because it is not in the HI QTL regions.

A4: Thank you for the suggestion. Since the major gene of maize HI has been cloned during our revision of this manuscript. Relevant discussions were added.

Q5. Figure 4b/c/d are unclear. Figure 4b shows most reads at higher levels than the diploid line the haploid embryo. The sperm in 4c shows several copy number gains. The embryo then shows copy number gain at the sites that were euploid in the sperm and copy number loss at the sites that were gained in the sperm. Similarly in figure 4d the sperm has a large loss in copy number that is gained in the embryo. Can the authors please clarify what is being represented in the figure? A standard copy number plot showing the number of genome copies at each location would be easier to follow, especially when switching between triploid, haploid, and diploid

genomes.

A5: We are sorry for the confusing due to the unclear legend of Fig. 4. In each panel of Fig. 4, the upper dot plot represents CNV by the $\log_2(\text{CNR})$ value, CNV is presumed if the continuous and dispersed obviously $\log_2(\text{CNR})$ values between large segments (> 10 Mb) are observed. The lower dot plot represents the genetic background which means paternal genome deletion if the value is greater than 0, in contrast, maternal genome deletion if the value is less than 0. The upper and lower figure of each panel is referring to the same sample. Fig. 4b means a haploid embryo, which is euploid (the upper dot plot shows) but loses the whole paternal genome (the lower dot plot shows). Similarly, in Fig. 4a, c-f, ploidy and genetic background was shown for each embryo (or endosperm). Clarification was added in the legend.

REVIEWERS' COMMENTS:

Reviewer #1 (Remarks to the Author):

This study by Li et al., is a nice complement to three recent publications (ref. 26, 27, 28 from the revised manuscript) that have solved the long held mystery behind the genetic basis of in vivo haploid induction in Stock -6 based inducers in Maize. All of them point to a sperm specific phospholipase A, a major gene QTL that governs the haploid induction. This study by Li et al., is a good supplement to those studies wherein the current study sheds light on the mechanistic basis behind this interesting phenomena which was not addressed in those publications. Here, they have exploited single cell sequencing approach to provide support to their hypothesis that chromosome fragmentation initiated in sperm nuclei during haploid mitotic divisions is a major cause for its subsequent elimination during embryonic divisions after fertilization. However, the question of how a mutation in a sperm specific phospholipase induces chromosome fragmentation still remains open. It will be interesting to see in future how a mutation in a phospholipase can trigger chromosome fragmentation.

While the authors have satisfactorily addressed most of the reviewers comments of the earlier version manuscript, still few clarifications and edits are necessary.

1. Which is the correct nomenclature CHOI3 (Candidate High Oil Inducer 3) or CHOI_3 . The latter one with subscript 3 appears like referring a molecular formula of a chemical. I believe as per the ref no: 30 cited in the manuscript the first one is the proper way of referring the strain.
2. Expand all abbreviations when used at the first instance For eg. TTC in line 94, CHOI3, CAU etc as applicable
3. In order to maintain the flow, lines 245-246 starting from "although we can haploid embryo." should follow after the line 249 (Table 1). The current flow indicates that their experimental results support the alternative possibility.
4. As pointed by other reviewers, though the manuscript is readable, still it requires grammatical fixes at many places. A thorough revisiting of entire manuscript for grammatical correction is recommended. Some examples are given below

Eg. Line 128-129 contain nature centromere sequences → containing centromere satellite sequences

Line 145 limit sample size should be limited sample size

Line 156 frequency of aneuploid should be frequency of aneuploidy

Line 159-160 seems be more serious → seems to be more severe?

Line 166 share a genetic background share a common genetic background?

Line 254 serious can be severe

Line 360 phosphoipid → phospholipid

Line 355-356, : Please remove " During the revision of this manuscript"

Line 650-651 "TTC dyed color" can be TTC staining

Line 680 difference of aneuplod ? aneuploid is spelled wrong through out the figure 3 legend.

Likewise, text legends in the supplementary information also need some attention for their grammar and sentence construction. An example is given below

Eg. Supplementary fig:1

a. "selfing ears and crossing ears in four lines" can be self pollinated and cross pollinated ears from four different genotypes/crosses.

b. "abortion kernels" should be aborted kernels

c. "In a and c, the bar" = 2.5cm simply put it as Scale bar = 2.5cm

Supplementary fig: 2

Symbol 1 to 4 represent kernels with abortion in the latter, middle, early stage of what?

Reviewer #2 (Remarks to the Author):

In this revision the authors have addressed my concerns satisfactorily. The findings reported here should be published to help address the mechanism of the stock 6 mutation (phospholipase ko) in genome stability and haploid induction. The manuscript's impact should be high.

I have a few minor suggestions. I do not need to examine the revision incorporating these changes and fixes.

Table 1: For clarity, pollen and sperm totals should be listed as separate columns and not as bracketed or not bracketed.

Figure 3: Total amount -> Total count.

Figure 3, 4 and 5. Ploid and aneuploid sperms (red and blue color) are not easy to discern. Use a different or lighter color, or no background color for the cytoplasm.

English style and grammar throughout the manuscript need careful editing and fixing. A few examples are listed below. There are many more that I do not have the time to address.

Line

89 weaker than of regular lines-> weaker than that of regular lines

143 The difference of aneuploidy frequency-> The difference in aneuploidy frequency

145 caused by the limit sample size.-> ...limited sample size.

149 chromosomes and five pollens containing two aneuploid sperm -> two aneuploid sperms

Reviewer #3 (Remarks to the Author):

The authors have met all my concerns. I believe the paper will have significant interest outside of the field of plant genomics, and is now suitable for publication in Nature Communications.

REVIEWERS' COMMENTS:

Reviewer #1 (Remarks to the Author):

This study by Li et al., is a nice complement to three recent publications (ref. 26, 27, 28 from the revised manuscript) that have solved the long held mystery behind the genetic basis of in vivo haploid induction in Stock -6 based inducers in Maize. All of them point to a sperm specific phospholipase A, a major gene QTL that governs the haploid induction. This study by Li et al., is a good supplement to those studies wherein the current study sheds light on the mechanistic basis behind this interesting phenomena which was not addressed in those publications. Here, they have exploited single cell sequencing approach to provide support to their hypothesis that chromosome fragmentation initiated in sperm nuclei during haploid mitotic divisions is a major cause for its subsequent elimination during embryonic divisions after fertilization. However, the question of how a mutation in a sperm specific phospholipase induces chromosome fragmentation still remains open. It will be interesting to see in future how a mutation in a phospholipase can trigger chromosome fragmentation.

We appreciated the nice comments and suggestions. It is our next plan to understand how the mutation of the phospholipase A gene can trigger chromosome fragmentation.

While the authors have satisfactorily addressed most of the reviewers comments of the earlier version manuscript, still few clarifications and edits are necessary.

Q1. Which is the correct nomenclature CHO13 (Candidate High Oil Inducer 3) or CHO₁₃. The latter one with subscript 3 appears like referring a molecular formula of a chemical. I believe as per the ref no: 30 cited in the manuscript the first one is the proper way of referring the strain.

A1. Thanks for your suggestion. We corrected them.

Q2. Expand all abbreviations when used at the first instance For eg. TTC in line 94, CHO13, CAU etc as applicable

A2. Thank you for the suggestion. We have added the full names for all the abbreviations.

Q3. In order to maintain the flow, lines 245-246 starting from “although we can haploid embryo. “ should follow after the line 249 (Table 1). The current flow indicates that their experimental results support the alternative possibility.

A3. We have reorganized this part. Thank you!

Q4. As pointed by other reviewers, though the manuscript is readable, still it requires grammatical fixes at many places. A thorough revisiting of entire manuscript for grammatical correction is recommended. Some examples are given below

Eg. Line 128-129 contain nature centromere sequences → containing centromere satellite sequences

Line 145 limit sample size should be limited sample size
Line 156 frequency of aneuploid should be frequency of aneuploidy
Line 159-160 seems be more serious → seems to be more severe?
Line 166 share a genetic background share a common genetic background?
Line 254 serious can be severe
Line 360 phospoipid → phospholipid
Line 355-356, : Please remove “ During the revision of this manuscript”
Line 650-651 “TTC dyed color” can be TTC staining
Line 680 difference of aneuploid ? aneuploid is spelled wrong through out the figure 3 legend.
Likewise, text legends in the supplementary information also need some attention for their grammar and sentence construction. An example is given below
Eg. Supplementary fig:1
a. “selfing ears and crossing ears in four lines” can be self pollinated and cross pollinated ears from four different genotypes/crosses.
b. “abortion kernels” should be aborted kernels
c. “In a and c, the bar” = 2.5cm simply put it as Scale bar = 2.5cm
Supplementary fig: 2
Symbol 1 to 4 represent kernels with abortion in the latter, middle, early stage of what?

A4. Thank you very much for the carefully check which is very helpful. We have corrected all of them and also double check the whole manuscript again.

Reviewer #2 (Remarks to the Author):

In this revision the authors have addressed my concerns satisfactorily. The findings reported here should be published to help address the mechanism of the stock 6 mutation (phospholipase ko) in genome stability and haploid induction. The manuscript's impact should be high.

Thanks for the nice comments.

I have a few minor suggestions. I do not need to examine the revision incorporating these changes and fixes.

Table 1: For clarity, pollen and sperm totals should be listed as separate columns and not as bracketed or not bracketed.

We have taken this suggestion and listed the pollen and sperm as separate columns

Figure 3: Total amount -> Total count.

We have corrected it.

Figure 3, 4 and 5. Ploid and aneuploid sperms (red and blue color) are not easy to discern. Use a different or lighter color, or no background color for the cytoplasm.

The color of euploid embryos and endosperms was changed into dark blue.

English style and grammar throughout the manuscript need careful editing and fixing. A few examples are listed below. There are many more that I do not have the time to address.

Line

89 weaker than of regular lines-> weaker than that of regular lines

143 The difference of aneuploidy frequency-> The difference in aneuploidy frequency

145 caused by the limit sample size.-> ...limited sample size.

149 chromosomes and five pollens containing two aneuploid sperm -> two aneuploid sperms

Thank you very much for the carefully check which is very helpful. We have corrected all of them and also double check the whole manuscript again.

Reviewer #3 (Remarks to the Author):

The authors have met all my concerns. I believe the paper will have significant interest outside of the field of plant genomics, and is now suitable for publication in Nature Communications.

Thank you!